DOI: 10.1038/s41467-018-04215-7　　**OPEN**

# Parental haplotype-specific single-cell transcriptomics reveal incomplete epigenetic reprogramming in human female germ cells

Ábel Vértesy [1,2], Wibowo Arindrarto [3], Matthias S. Roost[4], Björn Reinius[5], Vanessa Torrens-Juaneda[4], Monika Bialecka[4], Ioannis Moustakas[3,4], Yavuz Ariyurek[6], Ewart Kuijk[2], Hailiang Mei[3], Rickard Sandberg [5], Alexander van Oudenaarden[1,2] & Susana M. Chuva de Sousa Lopes [4,7]

In contrast to mouse, human female germ cells develop asynchronously. Germ cells transition to meiosis, erase genomic imprints, and reactivate the X chromosome. It is unknown if these events all appear asynchronously, and how they relate to each other. Here we combine exome sequencing of human fetal and maternal tissues with single-cell RNA-sequencing of five donors. We reconstruct full parental haplotypes and quantify changes in parental allele-specific expression, genome-wide. First we distinguish primordial germ cells (PGC), pre-meiotic, and meiotic transcriptional stages. Next we demonstrate that germ cells from various stages monoallelically express imprinted genes and confirm this by methylation patterns. Finally, we show that roughly 30% of the PGCs are still reactivating their inactive X chromosome and that this is related to transcriptional stage rather than fetal age. Altogether, we uncover the complexity and cell-to-cell heterogeneity of transcriptional and epigenetic remodeling in female human germ cells.

[1] Hubrecht Institute-KNAW (Royal Netherlands Academy of Arts and Sciences) and University Medical Center, 3584 CT Utrecht, The Netherlands. [2] Department of Genetics, Center for Molecular Medicine, Cancer Genomics Netherlands, University Medical Center Utrecht, Utrecht, 3584 CG, The Netherlands. [3] Department of Biomedical Data Sciences, Sequencing Analysis Support Core, Leiden University Medical Center, 2333 ZC Leiden, The Netherlands. [4] Department of Anatomy and Embryology, Leiden University Medical Center, 2333 ZC Leiden, The Netherlands. [5] Department of Cell and Molecular Biology, Karolinska Institutet, 171 77 Stockholm, Sweden. [6] Leiden Genome Technology Center, Leiden University Medical Center, 2333 ZC Leiden, The Netherlands. [7] Department of Reproductive Medicine, Ghent University Hospital, 9000 Ghent, Belgium. These authors contributed equally: Alexander van Oudenaarden, Susana M. Chuva de Sousa Lopes. Correspondence and requests for materials should be addressed to A.V.O. (email: a.vanoudenaarden@hubrecht.eu) or to S.M.C.d S.L. (email: Lopes@lumc.nl)

In the mammalian germline, the paternal and maternal epigenetic marks are removed to equalize the (epi)genome before meiotic entry. Key aspects of the epigenetic reprogramming in germ cells are the erasure of parent-specific genomic imprints and, in females, the reactivation of the inactive X chromosome. As a result, the expression of both imprinted and X-linked genes change from monoallelic to biallelic. In humans, the development of female germ cells, including the timing of meiotic entry, is strongly asynchronous[1–4] and several developmental stages ranging from early primordial germ cells (PGCs) to primordial follicles can be observed simultaneously in the same female gonad, from the second trimester onwards[4–6].

In recent years there has been major progress towards understanding the genetic[7, 8] and epigenetic regulation in fetal germ cells[9–11]. Pioneering work[4, 10] identified a pronounced transcriptional heterogeneity in human PGCs from week 11 onwards, using single-cell RNA sequencing. The authors identified heterozygous single nucleotide polymorphisms (SNPs) based on RNA sequencing data and concluded that X chromosome in PGCs was already reactivated in week 4 human embryos. This conclusion was based on the expression of a few selected genes, some being reported as XCI-escapees[12, 13]. Moreover, SNP calling from single-cell RNA sequencing is affected by low coverage, RNA modifications and it does not allow haplotype reconstruction. Without haplotyped chromosomes and good coverage of informative, non-escaping X-linked genes, allelic expression status of the X chromosome in humans remains elusive.

Here, we have combined high quality exome sequencing of fetal and maternal DNA samples with single cell RNA-sequencing of five donor (D) fetuses (Fig. 1). This allowed us to reconstruct the parental haplotypes of each of them. Thus, we were able to quantify the changes in chromosome-wide haplotypic expression. This revealed the dynamics the erasure of parent-specific genomic imprints and, in females, the reactivation of the inactive X chromosome.

## Results

**Germ cells cluster by stages of germ cell development**. First, we have noticed that the previously described heterogeneity is structured. Three distinct sub-populations were consistently present at specific locations in the human female gonad during developing (Supplementary Fig. 1). Human germ cells, homogenous during first trimester, progress to the second and third trimesters, by upregulating DDX4 and downregulating POU51F and PDPN; whereas most germ cells seem to express KIT (Supplementary Fig. 1a, b).

To determine whether the transcriptional signature of these distinct sub-populations remains unchanged during fetal development, we isolated and sequenced RNA from single cells from human fetal gonads (N = 73, including germ cells and gonadal somatic cells) and adrenal glands (N = 35 adrenal cells) from 8 to 14.4 weeks of development using SMART-seq2 technology[14]. We also incorporated previously published data from a total of 84 germ cells and 38 somatic cells from the additional 5 human female fetuses of 4–17 week donors (D7-D11)[10] that passed our quality control.

After alignment and quality control, a total of 155 female gonadal cells and somatic cells were classified by hierarchical clustering (Spearman correlation) based on the expression of 72 key genes covering the most relevant aspects of germ cell development ranging from primordial germ cell formation to their entry into meiosis involved, as well as several somatic genes[7, 9] (Fig. 2a and Supplementary Fig. 2). Somatic (N = 26) and germ cells (N = 129) clustered separately. Among the 129 germ cells, three distinct major clusters emerged 1) POU5F1⁺PDPN⁺ early stage PGCs (PGCs; N = 76) characterized by high levels of pluripotency and early germ cells markers, 2) pre-meiotic late germ cells (LGCs; N = 25) characterized by moderate levels of early, late, and meiotic germ cell markers, and 3) meiotic germ cells (MGCs; N = 28), characterized by high levels of meiotic markers (Fig. 2a).

Multidimensional scaling separated female germ cells by stage (PGC, LGC, MGC) in the first dimension and germ cells from somatic cells in the second dimension (Fig. 2b, c). This confirms that germ cells cluster by transcriptional program instead of by developmental age. Using Monocle[15], we computed the minimum spanning tree (Supplementary Fig. 3a) and ranked the germ cells along a pseudo developmental timeline (Fig. 2d). The list of genes enriched at least tenfold per stage, instead of age, provided a biologically meaningful marker gene list that characterizes each stage (PGC, LGC, MGC) (Supplementary Fig. 3b; Supplementary Data 1).

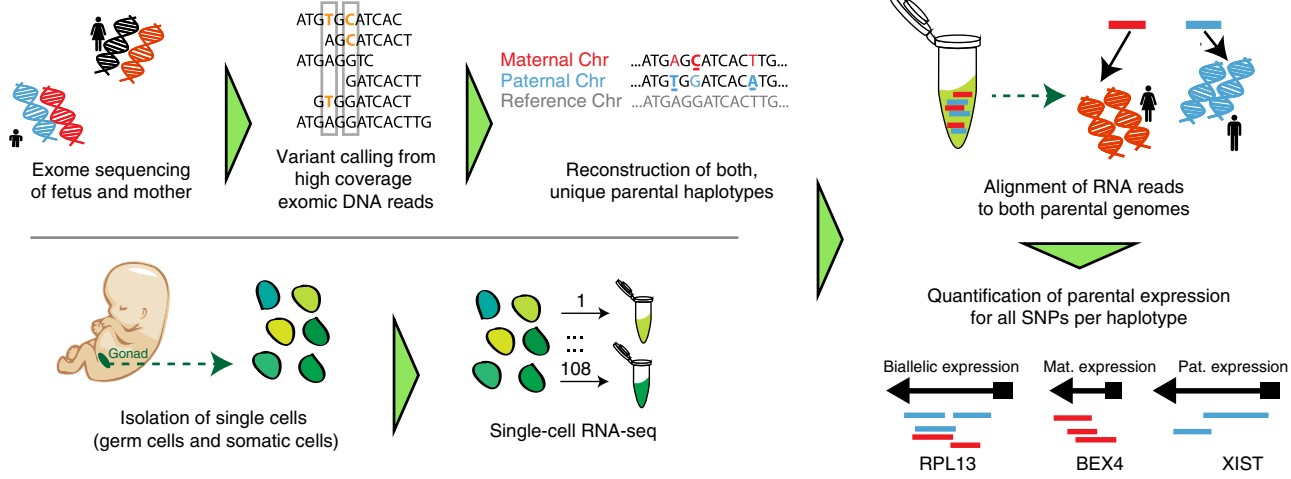

**Fig. 1** Parental haplotype reconstruction with single-cell sequencing detects genome wide allelic expression. The workflow combined high coverage exome sequencing of fetuses and mothers, used for variant calling to reconstruct the parental haplotypes for each fetus (using SNPs that are both heterozygous in the fetus and homozygous in the mother); isolation of single cells from the fetal gonad and adrenal gland, followed by RNA sequencing using Smart-seq2; and the alignment of the RNA reads per fetus to both parental genomes and the quantification of parental expression for all informative SNPs per haplotype

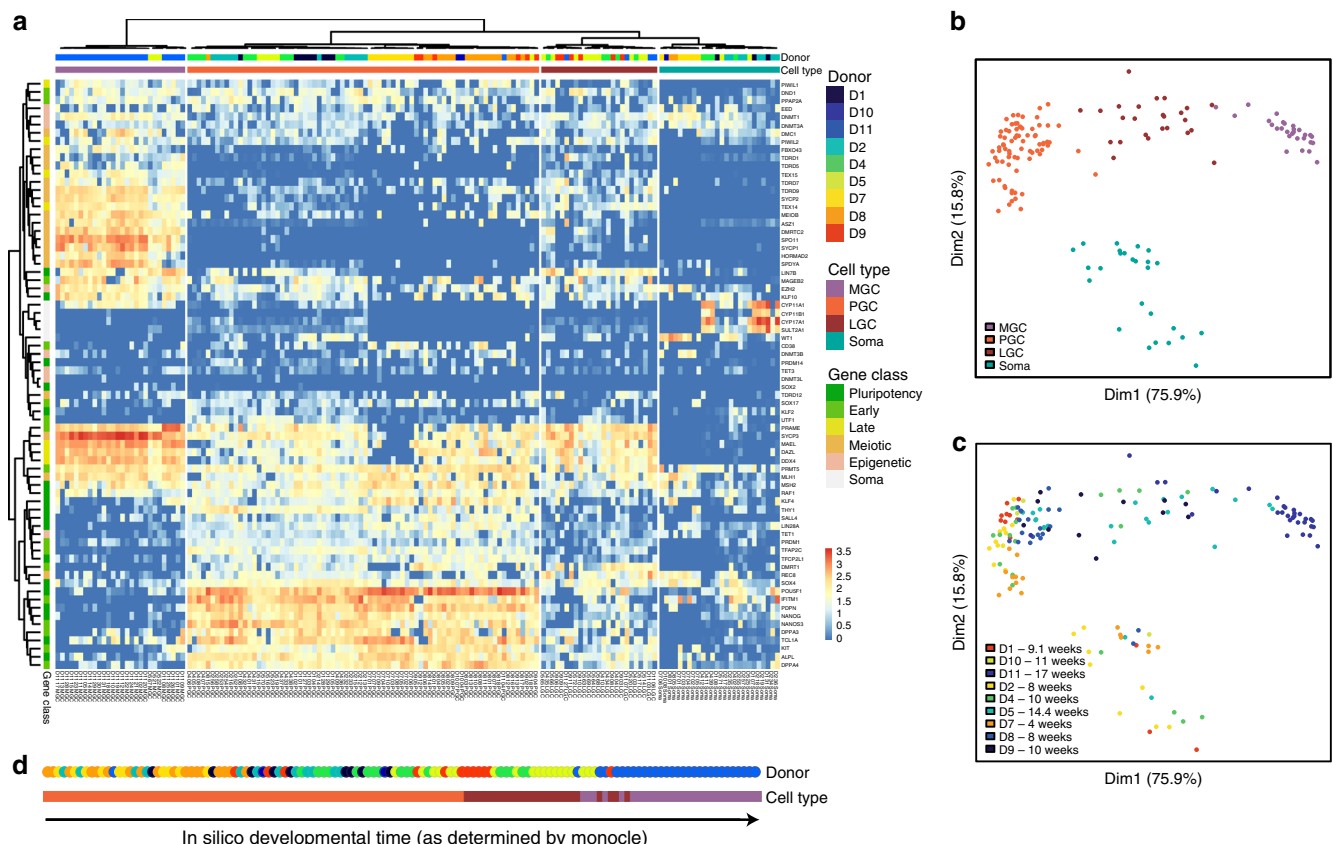

**Fig. 2** Fetal age does not determine developmental stage of human germ cells. **a** Unsupervised hierarchical clustering of single female human germ cells, and the associated gene expression heatmap of germline-specific genes, combining our dataset with female cells from Guo et al., 2015 yielded a total of 129 female germ cells and 26 female somatic cells from 9 different donors (D). The germ cells segregated into categories representing 3 different developmental stages, instead of segregating by donor or fetal age. The categories represent the transcriptional signatures of primordial germ cells (PGC), late germ cells (LGC) and meiotic germ cells (MGC). **b**, **c** Multidimensional scaling plots showing the individual somatic cells and germ cells (our dataset combined with that of Guo et al., 2015) color-coded by developmental stage (PGC, LGC, MGC) (**b**) and by donors of different fetal age (weeks) (**c**). **d** Individual germ cells (our dataset combined with that of Guo et al., 2015) ranked by their respective gene expression profiles using Monocle. This ranking is largely consistent with the independently identified developmental stages. Cells are colored according to the fetal age and developmental stage (PGC, LGC, MGC)

### Table 1 Basic characteristics of the fetal material used in the study

| Fetus basic characteristics | Fetus | | | Mother | | | SNP haplotypes of fetus | |
|---|---|---|---|---|---|---|---|---|
| ID Sex Age (weeks. days) | Median DNA coverage | HQ SNPs (×10⁶) | HQ het SNPs (0/1) (×10⁵) | Median DNA coverage | HQ SNPs (×10⁶) | HQ hom SNPs (0/0 and 1/1) (×10⁶) | Paternal (M:0/0 and F:0/1) | Maternal (M:1/1 and F:0/1) |
| D1 F 9.1 | 50 | 1.6 | 2.4 | 75 | 1.9 | 1.5 | 49,928 | 66,756 |
| D2 F 8.0 | 48 | 1.5 | 1.9 | 40 | 1 | 0.9 | 35,105 | 48,319 |
| D3 M 18.0 | 53 | 1.6 | 2.5 | 54 | 1.6 | 1.4 | 56,013 | 76,755 |
| D4 F 10.0 | 41 | 1.0 | 1.1 | 39 | 1 | 0.9 | 21,191 | 34,263 |
| D5 F 14.4 | 59 | 1.7 | 2.3 | 55 | 1.7 | 1.4 | 52,789 | 70,997 |

**Parental haplotypes show monoallelic expression per expected allele**. To date, epigenetic reprogramming in human germ cells has only been investigated by fetal age and not by germ cell stage[9, 10]. However, from our data, we hypothesized that the timing of epigenetic reprogramming may relate to the individual germ cell stage regardless of the developmental age of the fetus.

To study this in a rigorous manner, we analyzed the parent-specific expression of all SNP-containing X-linked genes and imprinted genes. We sequenced the exomes of the analyzed fetuses and their mothers and identified genes that contained heterozygous SNPs in the fetus, but were homozygous in the mother. Those SNPs allowed us to distinguish the parental origin

of the SNPs, to reconstruct the paternal and maternal haplotypes for each fetus (Table 1) and to quantify parental expression as opposed to simply monoallelic and biallelic expression of SNPs. After alignment and quality control, we excluded the cells in the lowest quintiles regarding the number of SNP-containing genes and allelic reads (Supplementary Fig. 3c) and analyzed further 53 female germ cells and 11 female somatic cells.

**Only imprinted genes in clusters show monoallelic expression**. From a list of confirmed human imprinted genes (compiled from refs.[16, 17] and www.geneimprint.com), we identified 39 that were

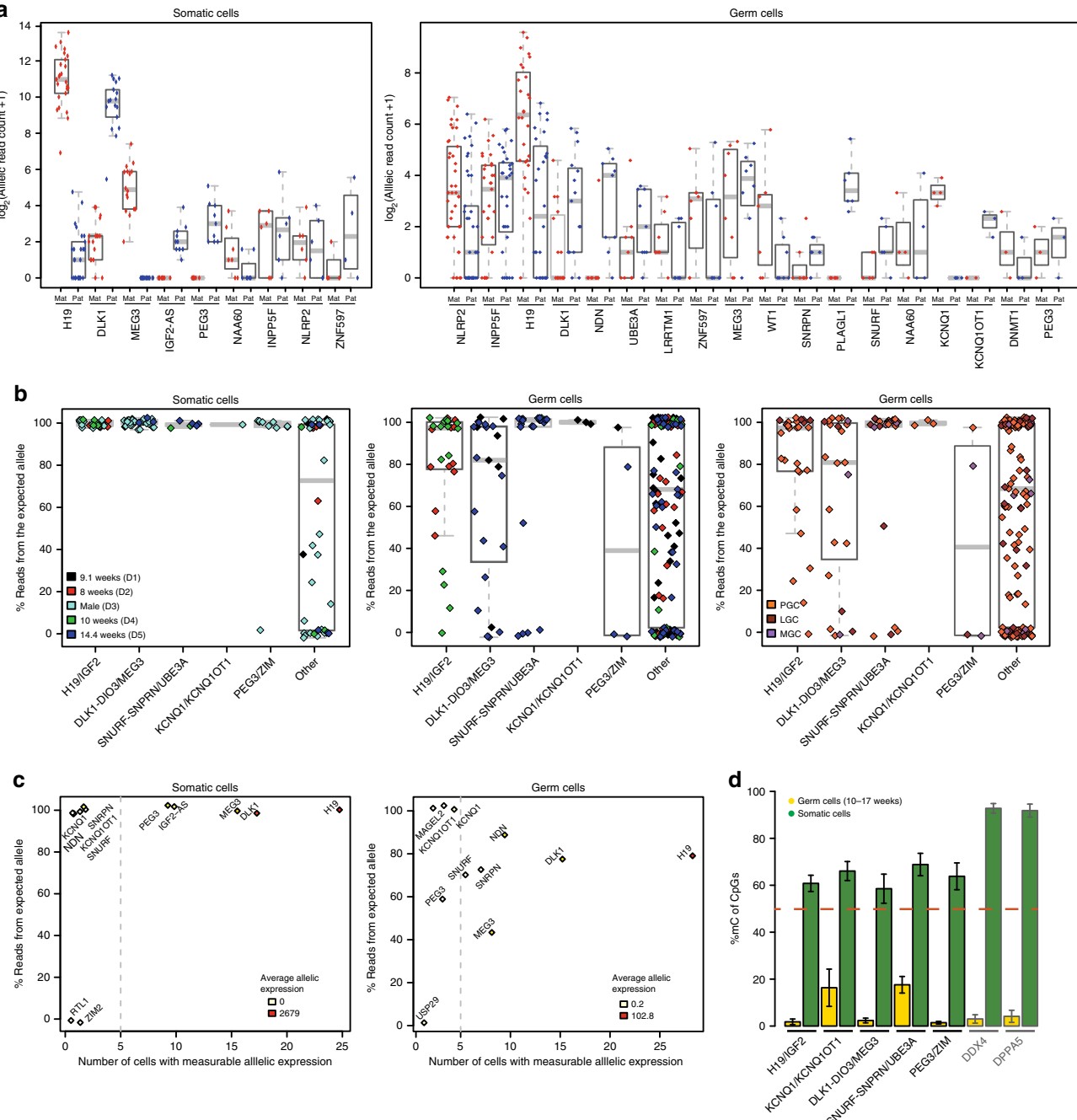

**Fig. 3** Parent-of-origin allele-specific expression of imprinted genes in human germ cells suggest cluster specific order of imprint erasure. **a** Expression of individual SNP-containing imprinted genes was separated in maternal and paternal read counts in somatic and germ cells. Gray bars depict the median. **b** Allele-specific expression of SNP-containing imprinted genes separated by imprinted gene-clusters in somatic cells (left panel) and germ cells (middle panel: colored by donor/age; and right panel: colored by developmental stage). Reported imprinted genes outside the five clusters altogether showed limited imprinting in soma and germline. Allelic read count ratios were plotted in relation to the expected (by imprinting) parental allele. Gray bars depict the median. **c** Average allelic bias (read count ratio) of genes from the 5 imprinted gene-clusters vs. the number of somatic (left) and germ cells (right) where allelic reads were available. The number of allelic read counts are denoted as the color of each dot (gene) from white to red. Gray line: monoallelic expression in 5 or more cells is significant for a specific gene being monoallelically expressed under the model of random allelic drop-out. **d** Quantification of DNA methylation in the analyzed and additional imprinting control regions in germ cells and somatic cells. The datasets used (germ cells and somatic cells) were from Guo et al., 2015. As control, the promoter region of DDX4 and DPPA5 shown to be methylated in somatic cells and demethylated in germ cells is also depicted. Extended analysis confirmed the variation in methylation patterns (Supplementary Figs. 3f, 4). Error bars denote the standard error of the mean

expressed and contained distinctive SNPs (Supplementary Data 2). We show the maternal and paternal allele-specific expression of each imprinted gene (expressed by at least 3 cells) in germ cells and somatic cells (Fig. 3a and Supplementary Data 2). From these, 14 out of 39 genes belonged to 5 well-established imprinted gene-clusters (*H19/IGF2*, *DLK1-DIO3/MEG3*, *KCNQ1/KCNQ1OT1*, *PEG3/ZIM*, and *SNURF-SNRPN/UBE3A*), each regulated by a differentially methylated imprinting control region (ICR)[18–20], whereas the remaining 25 (out of 39) genes are not reported to be part of ICR-regulated gene-clusters (Supplementary Data 2).

Interestingly, most of these 25 genes behaved less clearly as imprinted, showing partial biallelic expression, even in somatic cells (Fig. 3b). This may be due to their relatively low expression levels or alternatively because some of them may behave imprinted only in specific tissues. Baran et al., 2015[21] have reported variation in genomic imprinting across human individuals, tissues, and genes. Using their data, we showed that the 25 genes that belonged to the 5 imprinted gene-clusters were imprinted in a median of 97% of human tissues, compared to a median of 71% for the other 14 imprinted genes ($p = 0.028$, MWW test, one-sided) (Supplementary Fig. 3d). Moreover, the 14 genes from the 5 imprinted gene-clusters showed robust monoallelic expression from the expected allele in the somatic cells (Fig. 3b, c). The average allelic expression bias of these genes in germ cells was lower than in somatic cells ($p = 7e-04$, MWW, one-sided), but that also differed from the biallelic expression in autosomes ($p = 1e-07$, MWW, one-sided) (Fig. 3b, c and Supplementary Fig. 3e).

**Imprinting clusters determine erasure state**. In germ cells, genes of the *H19/IGF2*, *DLK1-DIO3/MEG3*, and *PEG3/ZIM* gene-clusters showed some degree of biallelic expression, but interestingly genes of the *KCNQ1/KCNQ1OT1* and *SNURF-SNRPN/UBE3A* (excluding *UBE3A* that is imprinted in the neuronal tissue only[22]) gene-clusters showed strong monoallelic expression from the expected allele (Fig. 3b, c). When the developmental stage of germ cells is depicted instead of the donor/age (Fig. 3b, right panel), even the LGCs and MGCs seem to keep monoallelic expression of imprinted genes from the expected allele, suggesting that 1) the imprint has not been erased yet or 2) the erasure of DNA methylation is complete, but this is not followed by biallelic expression and the observed monoallelic expression reflects 'left over' expression from earlier stages.

**DNA methylation supports monoallelic imprinted expression**. To distinguish between these two possibilities, we analyzed published data from 10–17 week female germ cells[10]. We found that the average rate of DNA methylation in the *KCNQ1/KCNQ1OT1* and *SNURF-SNRPN/UBE3A* (Prader–Willi syndrome (PWS)-ICR) ICRs was strikingly higher (about 18% methylated CpGs), when compared to the *H19/IGF2*, *DLK1-DIO3/MEG3*, and *PEG3/ZIM* ICRs in human germ cells, which showed complete loss of DNA methylation (Fig. 3d and Supplementary Fig. 3f). However, the DNA methylation in the *KCNQ1/KCNQ1OT1* and *SNURF-SNRPN/UBE3A* ICRs in the germ cells was considerably lower than the 60% methylation of imprinted ICRs in the somatic cells, suggesting ongoing demethylation. As control, we show that the promoter of *DDX4* and *DPPA5* are methylated in somatic cells and demethylated in germ cells (Fig. 3d).

In addition, we have FACS-sorted germ cells from 14 and 16 week female gonads, performed bisulfite conversion and sequenced the 5 ICRs of interest (Supplementary Fig. 4a, b). The percentage methylated and unmethylated reads was roughly 50:50 in somatic and maternal material, as expected. However, in the

FACS-sorted germ cells the DNA methylation in the *H19/IGF2*, *DLK1-DIO3/MEG3*, and *PEG3/ZIM* ICRs is clearly being erased, in contrast to the *KCNQ1/KCNQ1OT1* and *SNURF-SNRPN/UBE3A* ICRs that seem to retain the imprint. Informative SNPs present in the *H19/IGF2* (rs2071094 in the 16-week donor) and *PEG3/ZIM* (rs2302376 in the 14-week donor) ICRs confirmed that it is respectively the paternal and maternal allele that becomes demethylated in the germ cells. We concluded that even though PGCs seemed to initiate biallelic expression of both paternal and maternal imprinted genes, the timing of biallelic expression of imprinted genes occurs in a gene-cluster-specific manner during development. In the case of *H19*, the retained allelic expression in the PGCs was rather unexpected, particularly given the lack of DNA methylation in the *H19/IGF2*ICR. We verified the DNA methylation status of the proximal promoter region bordering with the transcriptional starting site (TSS) of *H19* (Supplementary Fig. 4c). Although the data suggested higher degree of demethylation in the PGCs compared with the somatic and maternal tissue, there were no informative SNPs in that region in the 2 donors analyzed to confirm demethylation in both alleles.

**29% of PGCs show a non-reactivated X chromosome**. Next, the allelic information was used to quantify the timing and dynamics of X chromosome reactivation at the single-cell level in PGCs, LGCs, and MGCs. To be able to pool the X-linked expression data, we assigned per single cell the presumably active X chromosome, based on the parental expression bias of the X chromosome (Fig. 4a). In our dataset, we identified 176 X-linked genes that were both expressed and contained distinctive SNPs (Supplementary Data 3). From those, 51 genes have been reported to escape X chromosome inactivation (XCI)[12, 13] and were analyzed independently. As expected, the XCI-escaping genes behaved similarly in somatic and germ cells, showing pronounced biallelic expression (Fig. 4b, top panel).

By analyzing the cells and genes pooled per cell state, both PGC and LGC + MGC showed biallelic expression of X-linked genes (X-proper) similar to that of autosomal genes and distinct from (monoallelically expressed) X-linked genes in somatic cells (Fig. 4b, bottom panel), suggesting X reactivation. However, we observed that some PGCs at 4–9 weeks showed a faint, but characteristic perinuclear spot of histone 3 lysine 27 trimethylation (H3K27me3), indicative of XCI, whereas this was not visible in germ cells at later stages until birth (Supplementary Fig. 5a-e) or in male germ cells (Supplementary Fig. 5f, g). This suggested that some PGCs still exhibited incomplete or ongoing X reactivation. Further analysis of other histone modifications (H3K9me3 and H3K4me2) failed to reveal further differences between male and female PGCs (Supplementary Fig. 5h, i).

To clarify this discrepancy, we sought to quantify more precisely the extent of XCI per single cell (Fig. 4c, Supplementary Fig. 6). To overcome random allelic dropout of individual genes, we added up all paternal and maternal reads for each chromosome per cell, respectively. This placed the X chromosome in the null-distribution of autosomes (Fig. 4c). Each individual donor has a unique set of SNPs, hence a wide range of X-reads, so we selected cells expressing at least 3 SNP-containing X-linked genes (non escapees) and a minimum of 33 reads. To avoid sequencing-depth (read count) related effects, we binned the chromosomes by expression level. For each bin, we calculated a confidence interval that contained 95% of the (biallelic) autosomes (Supplementary Fig. 6a). Unexpectedly, several PGCs (11 out of 38 PGCs, 29% of PGCs) contained X chromosomes that fell outside the biallelic interval, confirming incomplete or

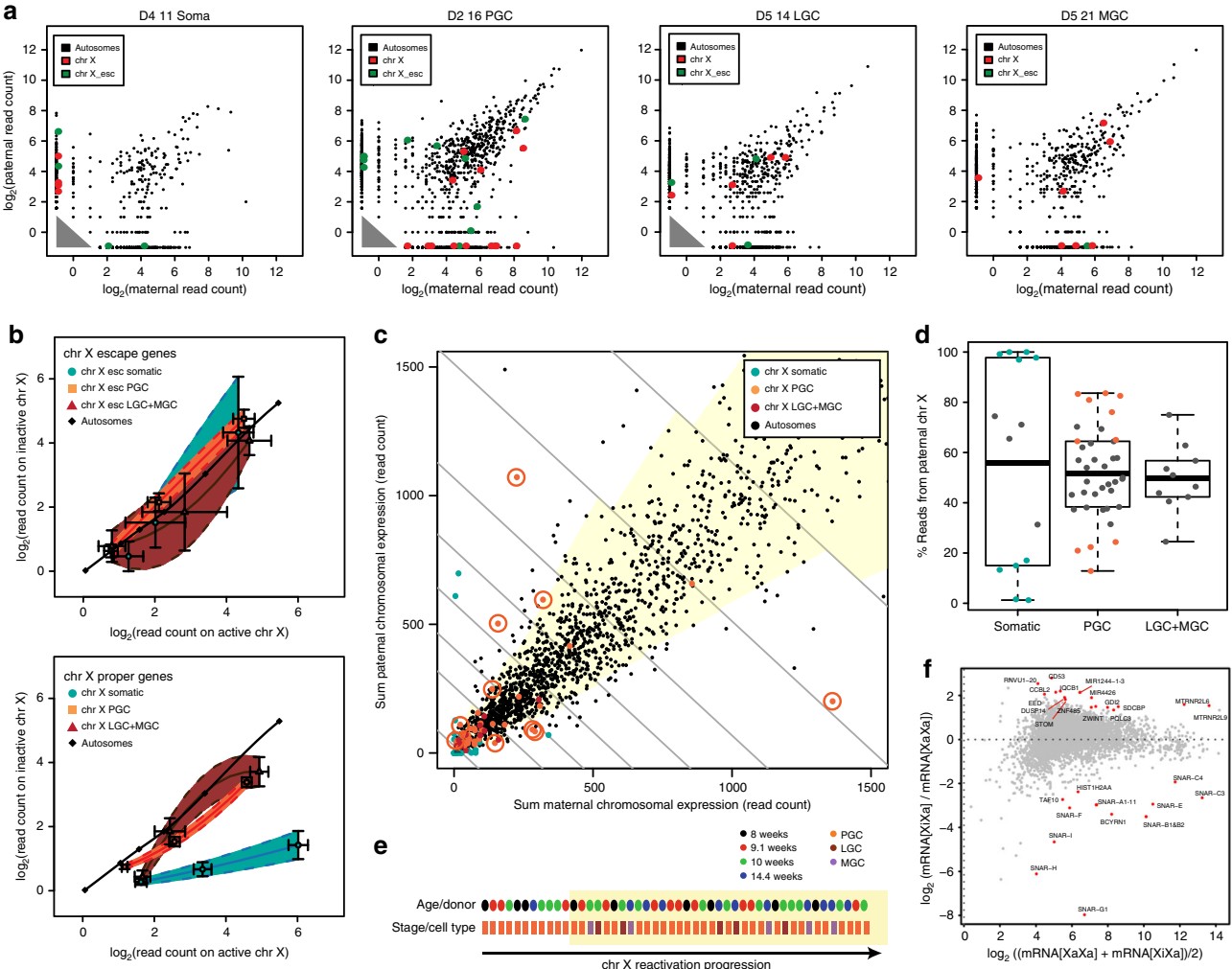

**Fig. 4** Parent-of-origin allele-specific expression of X-linked genes in human germ cells reveals incomplete X reactivation. **a** Examples of a somatic cell and germ cells at different developmental stages showing the maternal and paternal expression of SNP-containing autosomal genes (black dots), proper (non-escapee) X-linked genes (red dots) and X-linked genes know to escape inactivation (green dots). **b** Distribution of pooled allele-specific read counts of SNP-containing XCI-escaping genes (top) and X-proper genes (bottom) on the active and inactive X chromosome per cell type. The average ± standard error of the mean (SEM) is shown for each bin. **c** Allelic bias of individual autosomes (black) and sex chromosomes (colored by cell type and stage). The sum of maternal and paternal allele-specific read counts of SNP-containing genes per chromosome per single cell shows different degrees of reactivation. Chromosomes are binned per total allelic read counts (gray lines) to counter sequencing depth related technical effects. The yellow area in each bin is the 95% confidence interval as defined by allelic bias in autosomes. X chromosomes in orange circles are from germ cells that fall outside the 95% interval, therefore are significantly non-reactivated. Analysis pipeline depicted in Supplementary Fig. 6a and displayed in log-space in Supplementary Fig. 6b. The read counts of the X chromosome exclude reads from escapee genes. **d** Percentage of reads from the paternal X chromosome per cell type. Data points corresponding to cells that show significant allelic expression bias on the X chromosome (fell outside the 95% interval in Fig. 4c) are colored. The broken lines indicated the 25% quartiles. **e** Individual germ cells ranked according to their X chromosome expression bias (reactivation status). Cells are colored according to fetal age (top) and germ cell stage (bottom). X chromosomes in the yellow area are within the 95% confidence interval determined by the autosomes. **f** Differential gene expression between PGCs that contain X chromosomes with allelic expression bias comparable to autosomes (XaXa) and PGCs that are reactivating the silent X chromosome (XiXa). Red dots (with gene names) were significantly differentially expressed ($p < 0.05$). $P$ values were calculated using negative binomial distribution and corrected for multiple testing by the Benjamini-Hochberg method

ongoing X reactivation (Fig. 4c and log-scale in Supplementary Fig. 6b). Summarizing the same quantitative data in one dimension showed that the 11 PGCs ongoing reactivation (in orange) occupy extreme positions in the distribution of the X chromosome when considering allelic bias (Fig. 4d). PGCs showed either maternal expression bias (lower end of the distribution; 4 PGCs) or paternal expression bias (upper end of the distribution; 7 PGCs) (Fig. 4d), in accordance with the random nature of XCI. The somatic cells showed 5 cells with maternal and 5 cells with paternal expression bias (Fig. 4d). A similar outcome was reached using the geometrical mean

(GM, 99% confidence interval), a metric robust to outliers (Supplementary Fig. 6c). The significantly biased germ cells corresponded to less developed PGCs according to their monocle ranking (Fig. 4e and Supplementary Fig. 6d).

**XIST expression does not predict Chr X reactivation status.** Differences between mice and humans regarding *XIST* expression and its role regulating XCI have been reported in preimplantation embryos[23–25] and pluripotent stem cells[25–27]. In agreement, we also conclude that in human germ cells the expression levels of

*XIST* lack predictive value regarding the X-reactivation status (Supplementary Fig. 6e, f). This suggests that the molecular mechanism that regulates dosage compensation in humans is distinct from that in mice and that the major players remain to be identified. Recently, another X-linked long non-coding RNA *XACT* has been described as an important player in XCI in human pluripotent cells and preimplantation embryos[25]. *XACT* expression was not detected in any germ cells in our data set. We additionally provide a list of autosomal genes that showed positive or negative correlation in the X-reactivation ranked germ cells (Supplementary Fig. 7).

**Systematic analysis of published data confirms non-reactivation**. We then reanalyzed the XCI state of the germ cells here identified as PGCs from the Guo et al., 2015 single-cell transcriptomics dataset[10]. We found that 4 (*HDHD1, DMD1, USPX9, PLXNA3*) of the 5 representative X-linked genes presented to determine XCI state in that work were in fact reported escapees by Park et al., 2010[12]; and 2 of them (*HDHD1, USPX9*) also reported escapees by Zhang et al., 2013[13]. To analyze the allelic expression from the dataset in a more systematic manner, we first excluded all reported escapee genes and then we selected all expressed X-linked genes that harbored SNPs. From the 5 female individual donors, none were observed in PGCs of 4 week (D7) and 10 week (D9), but 8 week (D8) harbored 8, 11 week (D10) harbored 1 and 17 week (D11) harbored 7 SNP-containing expressed X-linked genes. Next, we calculated the median monoallelic bias of each gene per PGC. PGCs with at least 95% median monoallelic bias were considered in XCI state (Supplementary Fig. 6g). The systematic allelic expression analysis of the Guo et al., 2015 dataset supports our conclusions of incomplete X-reactivation, confirming that about 30% of PGCs (17 out of 57 PGCs) are still undergoing the process of X reactivation.

To investigate the genes that could be regulating X-reactivation in the PGCs, we compared the gene expression of the 11 PGCs undergoing X reactivation (XiXa) with the other PGCs (XaXa) (Fig. 4f and Supplementary Fig. 8). We identified several genes that were significantly differentially expressed ($p < 0.05$) in both groups, including the significant downregulation of *EED* in XaXa PGCs, which has also been reported in XaXa PGC in mice[28].

## Discussion

Concluding, we suggest caution when doing bulk analysis of human germ cells, because depending on the age of the human gonads, those will contain germ cells with several distinct transcriptional signatures, and that are in different phases of epigenetic reprogramming. Our data reveal surprising heterogeneity in the timing of memory erasure of allele-specific gene expression (imprinted genes and X-linked genes) during human germ cell development, both between individual loci and individual cells, thereby providing important information on the process and the timing of resetting the parental alleles in the human germ line.

## Methods

**Collection of human material**. The collection and use of human fetal tissues regarding the first and second trimester material was approved by the Medical Ethical Committee of the Leiden University Medical Center (P08.087). The human fetal material from first and second trimester was donated for research with informed consent from elective abortions without medical indication. The age of the fetuses was determined by obstetric ultrasonography (crown-rump length). For conversion between crown-rump length and developmental age please consult https://en.wikipedia.org/wiki/Crown-rump_length. Sex genotyping, using AMELOGENIN, was performed as described previously[6].

Paraffin sections from ovaries from the third trimester (perinatal death) were obtained from the tissue biobank, University Medical Center Utrecht, with approval from the Pathology Science Committee of the University Medical Center Utrecht (RP 2009-28).

**Immunofluorescence and confocal imaging**. Human gonads were dissected in saline solution (0.9% NaCl), fixed overnight in 4% paraformaldehyde, embedded in paraffin, sectioned and used for immunofluorescence as described previously[6]. Primary antibodies used were goat anti-DDX4 (1:1000, AF2030, R&D systems, Minneapolis, USA); rabbit anti-DDX4 (1:1000, ab13840, Abcam, Cambridge, UK); goat anti-POU5F1 (1:100, sc-8628, Santa Cruz Biotechnology, Dallas, USA); mouse anti-PDPN (1:100, ab77554, Abcam, Cambridge, UK); rabbit anti-KIT/CD117 (1:100, A450229, DAKO, Glostrup, Denmark); rabbit anti-H3K27me3 (1:500, 07-449, Millipore, Darmstadt, Germany); rabbit anti-H3K4me2 (1:500, 07-030, Millipore, Darmstadt, Germany); rabbit anti H4K9me3 (1:500, 07-442, Millipore, Darmstadt, Germany). The secondary antibodies used were donkey anti-goat Cy3 (1:200, 705-165-147, Jackson Immuno Research, West Groove, USA), donkey anti-rabbit Alexa Fluor 488 (1:500, A-21206, Thermo Fisher Scientific, Paisley, UK) and donkey anti-mouse Cy3 (1:200, 715-165-150, Jackson Immuno Research, West Groove, USA). Nuclei staining was performed with 4',6-diamidino-2-phenyl-indole (DAPI, Life Technologies, Paisley, UK). Imaging was performed on an inverted Leica TCS SP5 confocal laser-scanning microscope (Leica Microsystems, Mannheim, Germany), using Leica Application Suite Advanced Fluorescence (LAS AF) software version 1.8.0 (Leica Microsystems, Mannheim, Germany).

**Exome sequencing**. DNA isolation: DNA was isolated as previously described[29]. Exome sequencing was performed by BGI (www.bgi.com) following standard protocols and by LGTC (www.lgtc.nl). At the LGTC, the generation of the library was performed using the KAPA DNA Library Preparation Kits for Illumina following the instruction of the manufacturer (Kapa BioSystems). Briefly, DNA was sheared by Covaris, followed by end repair, A-tailing and adapter ligation to both ends. DNA was amplified by ligation-mediated PCR (LM-PCR) and purified by Agencourt Ampure beads. The exome was enriched by hybridization of the amplified library with SureSelect Human all exome (v3) probes following the instructions of the manufacturer (Agilent Technologies). The enriched samples were quantified on Agilent Bioanalyzer and sequenced on Illumina HiSeq2000 using v3 reagents generating 100 bp long, pair-end tags. The sequencing was performed following the instructions of the manufacturer (Illumina).

**Single-cell RNA-sequencing**. Single cell isolation from human organs: The tissue culture dishes to use for the single cell picking were first coated for a couple of minutes (min) at room temperature with 0.5% bovine serum albumin (BSA, Life Technologies, Carlsbad, USA) in PBS and then filled with Dulbecco's Modified Eagle Medium/F12 Nutrient mixture (DMEM/F12, Life Technologies, Paisley, UK). Human gonads and adrenals were placed individually in tissue culture dishes and the organs were mechanically disrupted with sharp needles so that single cells were released. Single cells were then manually picked under a stereo microscope (Zeiss, Sliedrecht, the Netherlands) using a pulled glass capillary in picking volumes of ±0.5 µl medium and transferred to ice-cold PCR tubes containing 2.0 µl lysis buffer (1.9 µl 0.2% TritonX-100 in water + 0.1 µl Recombinant RNase inhibitor (TaKaRa 40U/µl Ref. 2313A) per tube), snap-frozen on dry ice and stored at −80 °C.

Reverse transcription: Single-cell full-length cDNA libraries were prepared using the Smart-seq2 protocol[14]. Briefly, 2.1 µl priming buffer mix (1 µl 10 mM dNTPs and 1 µl 10 µM Smarter oligo-dT primer) was added to the cell lysate and incubated for 3 min at 72 °C in a thermal cycler, and then put on ice. Reverse transcription (RT) was performed by the addition of 5.6 µl Smart-Seq2 RT mix (0.5 µl SuperScriptII (Invitrogen Cat. 18064-014); 2 µl 5 × SuperScriptII buffer; 0.5 µl 100 mM DTT; 2 µl 5 M betaine; 0.1 µl 1 mM MgCl₂; 0.25 µl Recombinant RNase inhibitor (TaKaRa 40U/µl Ref. 2313A); 0.1 µl 100 µM LNA strand switch primer; 0.15 µl water, per reaction) and incubation (90 min 42 °C; 10 "strand-switch" cycles of (2 min 50 °C; 2 min 70 °C); 4 °C) in a thermal cycler.

cDNA amplification: cDNA amplification was performed by the addition of 15 µl of Smart-seq2 PCR mix [12.5 µl 2× KAPA HiFi HotStart ReadyMix (KAPA Biosystems Ref. KK2602); 0.25 µl 10 µM ISPCR primers; 2.25 µl water, per reaction] and incubation [3 min 98 °C; 19 cycles of (20 s (s) 98 °C; 15 s 67 °C; 6 min 72 °C); 5 min 70 °C; 4 °C] in a thermal cycler. The cDNA was purified using 0.7:1 volume of AMPure XP beads (Beckman Coulter, Ref. A63882) according to the manufacturer's protocol (No. PT5163-1). The purified cDNA for each cell was inspected on an Agilent 2100 Bioanalyzer to determine cDNA concentration and size distribution, using Agilent High Sensitivity DNA chips (Ref. 5067-4626).

Tagmentation and sequencing: Successful cDNA libraries were tagmented using the transposase Tn5[30]. 1ng of cDNA in 5 µl water was mixed with 15 µl tagmentation mix (1 µl of Tn5; 2 µl 10× TAPS MgCl₂ Tagmentation buffer; 5 µl 40% PEG8000; 7 µl water, per reaction) and incubated 8 min at 55 °C in a thermal cycler. Tn5 was inactivated and released from the DNA by the addition of 5 µl 0.2% SDS and 5 min incubation at room temperature. Sequence library amplification was performed using 5 µl Nextera XT Index primers (Illumina, Ref. 15032356) and 15 µl PCR mix [1 µl KAPA HiFi DNA polymerase (KAPA Biosystems Ref. KK202); 10 µl 5× KAPA HiFi buffer; 1.5 µl 10 mM dNTPs; 2.5 µl water, per reaction], and incubation [3 min 72 °C; 30 s 95 °C; 10 cycles of (10 s 95 °C; 30 s 55 °C; 30 s 72 °C); 5 min 72 °C; 4 °C] in a thermal cycler. Sequencing libraries were purified using 1:1 volume of AMPure XP beads according to the manufacturer's protocol), inspected on an Agilent 2100 Bioanalyzer, using Agilent High Sensitivity DNA chips and the DNA concentration was measured using a Qubit 2.0 Fluorometer (Invitrogen) with

the Qubit dsDNA High Sensitivity Assay kit (Molecular Probes, Ref. Q32854). Pools of samples for multiplexing, with unique Illumina barcode for each cell, were prepared according to the Nextera XT DNA Sample Preparation Guide (No. 15031942 page 46, Illumina). DNA sequencing was performed on an Illumina HiSeq2000 (43 bp single-end) and on Illumina NextSeq500 (75 bp single-end).

**Primary data analysis.** Exome Sequence Alignment: FASTQ files of individual sequencing batches were first run through the FastQC tool (version 0.10.1) to identify remaining adapter sequences and those were clipped using the Cutadapt tool (version 1.2). This was followed by a synchronizing step, using a custom Python script, to remove reads whose pair was discarded by cutadapt. Regardless of whether adapters are found or not, low quality bases were then trimmed using the sickle tool (version 1.3) on paired-end mode. The FASTQ files were then run through the BWA MEM aligner (version 0.7.10)[31] using the '-M -t 10' flag with an additional '-R' flag to set the read group and run name. The reference genome used was the Human hg19 genome from UCSC (https://genome.ucsc.edu) containing only the chromosomes 1 to 22, the mitochondrial genome and the sex chromosomes (X and Y). The resulting alignment was directly piped to the samtools tool (version 0.1.18) for compression into a BAM file, which was then sorted using the SortSam tool from the Picard suite (version 1.124), setting the stringency to lenient. This step was followed by a run with the MarkDuplicates with the 'REMOVE_-DUPLICATES' flag set to 'true', from the same Picard suite.

Exome Variant Calling: BAM files from different sequencing runs were analyzed using the GATK suite (version 3.2.2)[32, 33] following their best practices pipeline at the time of analysis. This was done using an in-house script written in the Scala programming language that runs on the GATK Queue engine. The script takes as input all BAM files from all sequencing batches and several annotation files. It implements the GATK best practices pipeline, which starts from the indel realignment step, followed by base recalibration, and then a merge per sample of the input BAM files. The merged BAM files were then run though the GATK HaplotypeCaller tool with the scatter-gather option, followed by separate variant recalibrator steps on the SNPs and indels. The resulting VCF files were then filtered for high-quality heterozygous SNPs in the non-parental samples. The final list of variants as annotated using the SnpEff tool (version 4.0), setting the source genome to 'hg19'.

Construction of parental genomes: High quality (PASS filter tag), heterozygous SNPs from each fetus and high quality homozygous SNPs of the corresponding mother were selected, to generate both parental genomes from hg19. Low quality SNPs, indels, and maternally heterozygous SNPs were discarded. From SNPs that were heterozygous in the fetus, maternal homozygous reference-SNPs were utilized to generate the paternal genome; whereas maternal homozygous alternative-SNPs were incorporated into the maternal genome. Finally, we had individualized maternal and paternal genomes for each of the fetus analyzed.

Generation of transcriptome reference: From the individualized hg19 genomes, transcript sequences were extracted, as described[34]. The transcriptome reference was built using the RefSeq gene models for hg19 in UCSC. All isoforms derived of the same gene were merged, yielding 23,738 unique transcripts on the chromosomes 1 to 22, the mitochondrial genome and the sex chromosomes (X and Y). Coordinates of these transcripts were then used to generate individualized transcriptome references for each sample based on the individualized genome sequences generated (as described earlier).

Allele-specific alignment of single cell mRNA libraries: FASTQ files of the sequenced transcripts were run through FastQC tool (version 0.10.1), Cutadapt tool (version 1.2) and Trim Galore (version 0.4.1) to remove transposase sequence remainders (CTGTCTCTTATACAC). Files then were aligned against the generated transcriptome reference using BWA 6.2 with default parameters[35]. We only considered uniquely mapped reads as defined by a single optimal alignment (X0_tag == "X0:i:1"). We mapped each cell's library twice. Once against the maternal, and once against the paternal transcriptome references, we selected reads that mapped uniquely against both. If a read had smaller edit distance to one of the genomes, we called that the parental allele of origin. For allelic expression analysis, we worked with read counts (as opposed to TPM), to be able to correct for sampling effects and allelic dropout.

Gene expression measurements. Gene expression levels were measured using the RSEM software (version 1.2.16) RSEM was run on a merged FASTQ file for each sample, setting the '--bowtie2' flag and using Bowtie version 2.1.0, the '--num-threads' flag to 8, and '--calc-ci' flag. The RSEM reference was prepared using the rsem-prepare-reference executable with the same hg19 genome file as was used in the variant calling step and the UCSC RefSeq GTF file, with the rRNA and tRNA regions removed. For non-allelic transcriptome analysis, we used RSEM's Transcript Per Million (TPM) tables; to compare with literature, we used Fragment Per Kilobase Mapped reads (FPKM) tables.

**Secondary data analysis.** Visualization: The data was visualized using the R packages gplots[36], ggplot2[37], and Markdown Reports[38].

Transcriptional analysis: Single cells from female germ cells our dataset and from Guo et al., 2015[10] (Gene Expression Omnibus (GEO) database GSE63818) with at least 7500 genes expressed were used for transcriptomics analysis (129 female germ cells and 26 female somatic cells). The data was primarily analyzed with R (version 3.2.2) and python (version 2.3). A list of 72 genes of interest was

used for clustering by multidimensional scaling using the base R package stats. Spearman correlation based on those 72 genes of interest was performed using the pheatmap package[39]. FPKM values for each cell were median-normalized and used for Spearman correlation based on the expression of the same set of genes of interest. For the tenfold enrichment analysis, from the pooled normalized data sets, cells were grouped either by germ cell stage or by fetal age, and per group the mean expression of each gene was calculated. Based on those means, all the genes of one group that had tenfold higher expression levels compared to mean of the other groups combined were determined (pair-wise analysis). The complete list of enriched genes by fetal age and by germ cell stage (Supplementary Data 1).

Monocle analysis: To rank the 129 female germ cells by their maturation status, we utilized the R implementation of Monocle[15] in Bioconductor. We selected genes with >5 TPM in at least 4 cells. That set of genes (13,793 genes) was intersected with the set of 72 genes of interest, reducing the list of genes to use in monocle to 66 genes (SOX2; HORMAD2; TDRD5; DNMT3L; CYP11B1; SULT2A1 were not expressed in the Guo dataset and thus excluded). We confirmed that our dataset is close to log-normal and proceeded with the analysis ranking cells by Monocle's minimum spanning tree and pseudotime.

Filtering of single cells for allele-specific analysis: we filtered further on allelic information, excluding cells from the lowest quintile regarding allelic reads or number of SNP-containing genes (53 female germ cells and 11 female somatic cells from our dataset).

Imprinting analysis: SNP-containing expressed genes were categorized as "confirmed imprinted" only if denoted as "imprinted paternal" or "imprinted maternal" (Refs.[16, 17] and geneimprint.com) (Supplementary Data 2). Other categories including provisional data, predicted, isoform dependent, reported to be imprinted, conflicting data, and no reports of imprinting status (orthologue to an imprinted gene in mice) were excluded from further analysis, as analysis showed that SNP-containing expressed genes from this "predicted imprinted" gene list do not appear to be imprinted in the somatic cells. We used the latest HUGO annotation for all genes not found in our dataset.

X inactivation analysis: Only cells with at least 3 SNP-containing expressed proper X-linked genes were considered for analysis. SNP-containing expressed genes that were reported to escape XCI in both following studies[12, 13] were analyzed separately (Supplementary Data 3). For each the X-linked gene, the allele with the most read counts was considered "active". Boundaries of bins were determined such that transcripts are equally distributed per bin; except in Fig. 4b where the smallest sample pool, the LGC+MGC, determined bin-size (3 bins) for X-linked genes in PGC and somatic cells. Analysis of differential expression between the 11 PGC undergoing X reactivation (XiXa) and the other PGC (XaXa) was represented and significance considered at $p < 0.05$. For the analysis of the X chromosome expression bias in the PGCs of the Guo et al., 2015[10] dataset, the expressed X-linked genes (excluding escapees) that harbored SNPs per donor were identified. After that, the median of allelic bias per PGC in each donor was plotted.

DNA methylation external data sets: From DNA methylation data (.bed files) from the GEO database GSE63818[10], we extracted genomic coordinates for several imprinting control regions of H19/IGF2 (chr11: 2,021,070-2,021,302), KCNQ1/KCNQ1OT1 (chr11: 2,721,173-2,721,297), DLK1-DIO3/MEG3 (chr14: 101,292,152-101,292,376), SNURF-SNRPN/UBE3A (chr15: 25,200,010-25,200,249), and PEG3 (chr19: 57,351,942-57,352,097)[40] and the promoter region of DDX4 (chr5: 55,029,104-55,029,220) and DPPA5 (chr6: 74,063,525-74,063,669)[10]. Next, we calculated the average methylation rates of CpG-context cytosines (± standard deviation). The methylation datasets pooled for "germ cells" were four female KIT-positive germ cell samples (10 -week replica1, 10-week replica2, 11-week, 17-week) and for "somatic cells" were eight somatic samples (5-week brain, 5-week heart, KIT-negative sorted gonadal somatic cells from 7, 10, 11, 19-week male and 17-week female).

**FACS sorting and DNA methylation analysis.** FACS sorting: Human gonads from two individual female foetuses (14 and 16 week) were dissociated to single cell suspension using 1 mg/ml Collagenase II (Thermo Fisher, 17101015) in 0.25% Trypsin EDTA (Thermo Fisher, 25200056) overnight at 4 °C. The cells were re-suspended in 1% BSA in PBS and immunostained with EPCAM-VioBlue (1:100, 130-098-092, Miltenyi Biotech), TNAP-488 (1:100, 561495, BD Biosciences) and KIT-APC (1:400, 550412, BD Biosciences) for 45 min at 4 °C in the dark. Thereafter, the cells were re-suspended in FACS buffer (1% BSA and 2 mM EDTA in PBS) and prior to FACS-sorting, 7AAD (1:100, 420403, ITK Diagnostics) was added. Cell sorting was performed on BD FACSAria III (BD Biosciences, Erembodegem, Belgium) using BD Diva 8.0.2 software. Sorted cells were snap frozen and stored at -80 °C before further analysis.

DNA bisulfite conversion: Sorted germ cells, somatic and maternal tissue were lysed in 1 mg/ml Proteinase K (Qiagen, 19131) in 10 mM Tris-HCl (pH 8.5) for 1 h at 56 °C and directly used for bisulfite conversion using EZ DNA Methylation-Lightning kit (Zymo Research, D5031) according to manufacturer instructions. Regions of interest were amplified from bisulfite converted DNA (bsDNA) template using Platinum-Taq DNA polymerase (Invitrogen, 10966018) with tailed locus specific primers from the different ICRs[40] (with adaptors for sequencing) and the proximal promoter containing the transcriptional starting site (TSS) of H19 (Supplementary Table 1) and the following PCR parameters: 5 min 96 °C; 50 cycles of (45 s 94 °C; 45 s 58 °C; 45 s 72 °C); 10 min 72 °C; 4 °C.

NGS sequencing: Samples were barcoded and the amplicons converted to Illumina-compatible NGS libraries and sequencing was done using the MiSeq (Illumina) with the 600 bp v3 reagents kit following the manufacturers instructions. Using Illumina's pipeline (bcl2fastq.2.17.4) the fastq files for the individual samples were generated.

DNA methylation analysis: Data was cleaned from adaptor sequences and a minimum of a read length (50) and base quality (15) was ensured using Cutadapt v1.9.1[41] with settings: -q 15 --minimum-length 50. The paired read sequences were merged using FLASH v1.2.11[42] with default settings. Next, the merged sequences were aligned to the bisulfite converted genome using Bismark v0.18.2 with settings: --ambiguous -N 1 -p 3[43]. Bismark methylation extractor was then used to get the methylation pattern per read. Custom scripts were used to count the reads with different methylation patterns and to separate the aligned reads based on the SNP allele (frequency >5%) and put them in separate alignment files. Visualization of methylated CpGs in the regions of interest was performed based on Tabsat v1.0.2[44].

Cloning bisulfite-amplicons: To analyze the methylation in the H19 TSS region, these amplicons were cloned using the TOPO TA Cloning kit Dual Promoter (vector pCRTMII-TOPO and TOP10 competent bacteria) (Invitrogen) following the instructions of the manufacturer. The obtained colonies were screened by PCR and positive colonies analyzed by Sanger sequencing.

**Data availability**. RNA-seq data are deposited in Gene Expression Omnibus, accession number GSE79280. The analysis pipelines are available on https://github.com/vertesy/X-Reactivation under GNU GPLv3 license and on https://github.com/johnmous/methylationScripts.

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

## Acknowledgements

We would like to thank the Centre for Contraception, Abortion, and Sexuality (CASA) in Leiden and Den Hague for the collection of the human fetal material and P. Nikkels from the biobank of the University Medical Center Utrecht; the sequencing facilities from the LUMC (LGTC), the Hubrecht Institute and the Karolinska Institute for sequencing and primary data analysis; BGI Hong Kong and LGTC for exome sequencing and primary analysis. L. van Iperen for collecting the material, histology and help with immunostaining, M. Gomes Fernandes for sex genotyping, A. Melo Bernardo and A. de Graaf for help with confocal imaging, N. He and C. Pacios for testing primers, W. Reik, B. Spanjaard, and S. Dey for insightful discussions and advice and M. Sen for critical reading of the manuscript. This work was supported by the European Research Council Advanced (ERC-AdG-294325-GeneNoiseControl) to A.v.O.; the Bontius Stichting [PANCREAS] to M.S.R.; the Swedish Foundation for Strategic Research (FFL4) to R.S.; and the European Research Council Consolidator (ERC-CoG-725722-OVOGROWTH) to I.M., V.T.J., and S.M.d.C.S.L.

## Author contribution

A.V., A.v.O., S.M.d.C.S.L. conceived the study and designed the experiments; S.M.d.C.S. L. and M.S.R. performed the isolation of the human gonads and adrenals, performed the

single cell collection, isolated genomic DNA; E.K. and S.M.d.C.S.L. did immunostainings, imaging, and image analysis; B.R. and R.S. prepared single-cell RNA-seq libraries; A.V., Y.A., and B.R. performed sequencing; A.V., W.A. and H.M. performed sequence alignment and W.A. did variant calling; M.B., Y.A., I.M., and V.T.J. performed FACS-sorting of germ cells, bisulfite conversion, cloning, sequencing, and alignment; and AV developed the analysis of allele-specific expression and performed computations. All coauthors analyzed the data and wrote the manuscript. All authors approved the final version of the manuscript.

## Additional information

**Competing interests:** The authors declare no competing interests.

