## [Peer Review File · Nature Communications]

Reviewers' comments:

Reviewer #2 (Remarks to the Author):

The authors have done a good job at responding to the reviewer's comments. In this reviewer's opinion, this paper is a valuable study that provides a new point of view on human PGC reprogramming with a focus on X chromosome and imprint reactivation. The value of this work is that the authors use RNA-Seq and exome sequencing to better reveal biallelic and mono allelic expression in single PGCs rather than making assumptions based on just SNP data in RNA-Seq. The study performs a more focused analysis on a more accurately curated set of X-linked genes (both escapees and those that are proven subject to XCI) and finds the opposite result of the Guo paper, even using the Guo data. Future studies will be important to better define X-chromosome reactivation, which are beyond the scope of this paper.

Small grammatical errors were detected, that should be corrected.

Review – page 170 used EGC rather than PGC.

Page 198 – XACT expression was..... in (not on) our data set.

In the Figures where you are referring to a "D" (For example Fig 4a "D4,, D2 etc.) I am presuming that it is donor? It would be more helpful to define "D" in the figure legend as my first thought was it was developmental day and was confused.

Reviewer #3 (Remarks to the Author):

In their manuscript the authors addressed many of the points raised by the reviewers of the originally submitted manuscript. The manuscript has gained in clarity as well.

An important issue which remains mechanistically unanswered concerns the imprinted, allele-specific, gene expression that persists in the week 10-17 PGCs. Specifically, for most of the genes it is unclear whether the allelic expression in the PGCs could be due to a partial non-erasure of the methylation imprints, or whether another explanation is more likely. Given the low transcript numbers that were detected in the PGC cells, indeed, one alternative possibility is that these allelic transcripts are in part retained in the differentiating PGCs and do not reflect transcription at the stages studied.

The H19 gene is one of the few genes that showed allelic transcripts in the PGCs and yet, at the IGF2-H19 ICR there is an almost complete lack of DNA methylation in the week 10-17 PGCs (indicative of erasure of the ICR DNA methylation imprint). However, this methylation

data set originated from the Guo et al. study, which makes it difficult to assess its quality in the context of the current study and whether these data can indeed be integrated (same cells studied?, comparable developmental stages?, etc.)

In fact, the only cases for which partially retained DNA methylation at the ICR could possibly explain retention of allelic transcription are SNRPN/SNRF and KCNQ1OT1. However, here it is not clear whether in a subset cells the methylation imprints are maintained, or whether in all the cells there is a partial loss of DNA methylation in the weeks 10-17 PGCs. But, again here, the methylation data were retrieved from the earlier study by Guo et al. , which makes it difficult to know whether things are indeed fully comparable.

To address the extent of the apparent delayed erase of methylation imprints at imprinted loci, and whether this explains the observed allelic transcripts at several imprinted genes in PGCs, it would be desirable to include DNA methylation data. Imprinting erasure is all about removal of the allelic DNA methylation at ICRs, and no methylation data were generated on the germ cells that were developmentally staged and studied in the current work. Inclusion of focused methylation data would greatly enhance the interest of the manuscript.

Specific points:

- 1) For the KCNQ1OT1 ICR and the SNRPN ICR, could the authors provide data on methylation levels in their own PGCs, analyzed according to their temporal and phenotypic and developmental staging?
- 2) As an additional step towards understanding the apparent delayed erasure of imprinting, it would be advisable to generate bisulphite sequencing profiles on individual chromosomes at the KCNQ1OT1 and the SNRPN ICRs. Importantly, such data would indicate whether cellular heterogeneity or partial removal of DNA methylation in all the PGCs.
- 3) The retained allelic expression of H19 in the PGCs is rather unexpected, particularly given the lack of DNA methylation at this locus' ICR that was observed in the Guo et al. study. Importantly, the authors should verify the methylation status of the H19 promoter in the week 17-20 PGCs.

Minor points:

-In Figure 3a, the gene names in black indicate 'ICR controlled genes' whereas red gene names indicate genes 'not controlled by ICRs'. This distinction seems confusing. Almost all the genes shown are not directly controlled by ICRs.

-The use of both the terms IC and ICR is confusing. The latter term is used most often in the field.

Point-by-point answer to the reviewers

Reviewer #2 (Remarks to the Author):

The authors have done a good job at responding to the reviewer's comments. In this reviewer's opinion, this paper is a valuable study that provides a new point of view on human PGC reprogramming with a focus on X chromosome and imprint reactivation. The value of this work is that the authors use RNA-Seq and exome sequencing to better reveal biallelic and mono allelic expression in single PGCs rather than making assumptions based on just SNP data in RNA-Seq. The study performs a more focused analysis on a more accurately curated set of X-linked genes (both escapees and those that are proven subject to XCI) and finds the opposite result of the Guo paper, even using the Guo data. Future studies will be important to better define X-chromosome reactivation, which are beyond the scope of this paper.

We thank the reviewer for the positive comments.

Small grammatical errors were detected, that should be corrected.
The grammatical errors that we could detect have been corrected.

Review – page 170 used EGC rather than PGC.

We have modified the text as suggested.

Page 198 – XACT expression was..... in (not on) our data set.

We have modified the text as suggested.

In the Figures where you are referring to a “D” For example Fig 4a “D4,, D2 etc.) I am presuming that it is donor? It would be more helpful to define “D” in the figure legend as my first though was it was developmental day and was confused.

We have modified the text in the Figure legends as suggested.

Reviewer #3 (Remarks to the Author):

In their manuscript the authors addressed many of the points raised by the reviewers of the originally submitted manuscript. The manuscript has gained in clarity as well.

We thank the reviewer for the positive comments.

A.) An important issue which remains mechanistically unanswered concerns the imprinted, allele-specific, gene expression that persists in the week 10-17 PGCs. Specifically, for most of the genes it is unclear whether the allelic expression in the PGCs could be due to a partial non-erasure of the methylation imprints, or whether another explanation is more likely. Given the low transcript numbers that were detected in the PGC cells, indeed, one

alternative possibility is that these allelic transcripts are in part retained in the differentiating PGCs and do not reflect transcription at the stages studied.

We thank the reviewer to bring this matter to our attention. We performed additional experiments to clarify the issue raised and conclude that the monoallelic expression of genes from KvDMR and the SNRPN ICR in our FACS-sorted germ cells is indeed coincides with partial non-erasure of the methylation imprints. We have included these results in the new Supplementary Fig. S4.

B.) The H19 gene is one of the few genes that showed allelic transcripts in the PGCs and yet, at the IGF2-H19 ICR there is an almost complete lack of DNA methylation in the week 10-17 PGCs (indicative of erasure of the ICR DNA methylation imprint). However, this methylation data set originated from the Guo et al. study, which makes it difficult to assess its quality in the context of the current study and whether these data can indeed be integrated (same cells studied?, comparable developmental stages?, etc.)

With the inclusion of novel bisulfite data we confirmed that in our FACS-sorted PGCs that there is indeed erasure of the IGF2-H19 ICR DNA methylation imprint at 16wk and ongoing at 14wk. We have included those results in the new Supplementary Fig. S4.

C.) In fact, the only cases for which partially retained DNA methylation at the ICR could possibly explain retention of allelic transcription are SNRPN/SNRF and KCNQ1OT1. However, here it is not clear whether in a subset cells the methylation imprints are maintained, or whether in all the cells there is a partial loss of DNA methylation in the weeks 10-17 PGCs. But, again here, the methylation data were retrieved from the earlier study by Guo et al. , which makes it difficult to know whether things are indeed fully comparable.

This is an important point addressing the how the erasure of imprints happen. We analysed the fraction of methylated CpG-s in individual reads in all new bisulfite experiments. We found that majority (~70-75%) of all reads in the 14wk and 16wk germ cells for the SNRPN/SNRF and KCNQ1OT1 ICR are either methylated ($\geq 81\%$ or $\geq 84\%$ of CpG-s) or unmethylated ($\leq 16\%$ or $\leq 19\%$ of CpG-s), but there is a significant fraction of the reads that show partial methylation. The fractions of partially methylated reads were in most cases higher than that fraction observed in the maternal and somatic tissues. Taken together, these observations suggest that both cellular heterogeneity (coexistence of methylated an unmethylated states) and ongoing demethylation (partial loss of methylation marks) contribute to the population-level partial decrease in methylation. The detailed results of the additional experiments are included in Supplementary Fig. S4.

To address the extent of the apparent delayed erase of methylation imprints at imprinted loci, and whether this explains the observed allelic transcripts at several imprinted genes in PGCs, it would be desirable to include DNA methylation data. Imprinting erasure is all about removal of the allelic DNA methylation at ICRs, and no methylation data were generated on the germ cells that were developmentally staged and studied in the current work. Inclusion of focused methylation data would greatly enhance the interest of the manuscript.

We have included targeted bisulfite methylation data and presented the results in the new Supplementary Fig. S4. We found persistent methylation at the SNRPN/SNRF and KCNQ1OT1 ICRs up to 16wk, partial methylation at the H19 ICR up to 14wk, and the loss of methylation at the PEG3 and DLK1 ICRs.

Specific points:

1) For the KCNQ1OT1 ICR and the SNRPN ICR, could the authors provide data on methylation levels in their own PGCs, analyzed according to their temporal and phenotypic and developmental staging?

We have generated new data regarding the DNA methylation of the 5 ICR regions studied, including KCNQ1OT1 ICR and the SNRPN ICR, on FACS-sorted germ cells. We isolated germ cells according to their developmental staging in a new Supplementary Fig. S4. We isolated PGC, LGC and MGC populations in a marker gene based FACS strategy. The LGCs were too few to generate robust methylation data, but we show data on the FACS-sorted PGCs from 2 different donors and MGCs from the 16wk donor (the 14wk donor does not have MGCs yet).

2) As an additional step towards understanding the apparent delayed erasure of imprinting, it would be advisable to generate bisulphite sequencing profiles on individual chromosomes at the KCNQ1OT1 and the SNRPN ICRs. Importantly, such data would indicate whether cellular heterogeneity or partial removal of DNA methylation in all the PGCs.

We have generated bisulphite sequencing profiles at the KCNQ1OT1 and the SNRPN ICRs, but unfortunately in the two different donors obtained, there were no informative SNP in those two ICR regions. There were informative SNPs in the PEG (rs2302376) and H19 (rs2071094) ICR regions and there the expected allele was demethylated. Nevertheless, methylation patterns across individual reads for the KCNQ1OT1 and the SNRPN ICRs suggest that there is both partial removal of DNA methylation and coexistence of cells with distinct methylation status. (See answer to point C.) Overall our results indicate that there is partial retention of the imprint at the KCNQ1OT1 and the SNRPN ICRs and this in part can be

explained by an increased fraction of partially methylated reads (see right-most column in Fig.S4).

3) The retained allelic expression of H19 in the PGCs is rather unexpected, particularly given the lack of DNA methylation at this locus' ICR that was observed in the Guo et al. study. Importantly, the authors should verify the methylation status of the H19 promoter in the week 17-20 PGCs.

We agree with the reviewer that this observation is unexpected in light of the Guo et al data. Importantly however, the allelic expression from the H19 locus showed an intermediate bias, being distinct from the SNRPN/SNRF and KCNQ1OT1 loci. We now provide both the DNA methylation data on the H19 ICR as well as H19 proximal promoter/transcriptional starting site. Although the data suggests demethylation in the proximal promoter/TSS, the lack of informative SNPs in this region in the individuals analysed does not allow us to confirm if there is specific demethylation in the paternal allele.

The text reads:

“In the case of *H19*, the retained allelic expression in the PGCs was rather unexpected, particularly given the lack of DNA methylation in the H19/IGF2ICR. We verified the DNA methylation status of the proximal promoter region bordering with the transcriptional starting site (TSS) of *H19* (Supplementary Fig. 4c). Although the data suggested higher degree of demethylation in the PGCs compared with the somatic and maternal tissue, there were no informative SNPs in that region in the 2 donors analysed to confirm demethylation in both alleles.”

Minor points:

-In Figure 3a, the gene names in black indicate ‘ICR controlled genes’ whereas red gene names indicate genes ‘not controlled by ICRs’. This distinction seems confusing. Almost all the genes shown are not directly controlled by ICRs.

We have labeled all genes in Figure 3a in black as requested.

-The use of both the terms IC and ICR is confusing. The latter term is used most often in the field.

We have removed the term IC throughout the manuscript as requested.

REVIEWERS' COMMENTS:

Reviewer #3 (Remarks to the Author):

The authors addressed the remaining points of this reviewer.